# Agronomic and Physiological Traits Response of Three Tropical Sorghum (*Sorghum bicolor* L.) Cultivars to Drought and Salinity

Elvira Sari Dewi [1,2,*], Issaka Abdulai [1], Gennady Bracho-Mujica [1], Mercy Appiah [1] and Reimund P. Rötter [1,3]

1   Department of Crop Sciences, Tropical Plant Production and Agricultural System Modelling (TROPAGS), University of Göttingen, Grisebachstraße 6, 37077 Göttingen, Germany; iabdula@gwdg.de (I.A.); gennady.brachomujica@uni-goettingen.de (G.B.-M.); mercy.appiah@uni-goettingen.de (M.A.); reimund.roetter@uni-goettingen.de (R.P.R.)
2   Department of Agroecotechnology, Faculty of Agriculture, Universitas Malikussaleh, Aceh Utara 24355, Indonesia
3   Centre for Biodiversity and Sustainable Land Use (CBL), University of Göttingen, Büsgenweg 1, 37077 Göttingen, Germany
*   Correspondence: elvira@unimal.ac.id or elvirasari.dewi@stud.uni-goettingen.de

**Abstract:** Sorghum holds the potential for enhancing food security, yet the impact of the interplay of water stress and salinity on its growth and productivity remains unclear. To address this, we studied how drought and salinity affect physiological traits, water use, biomass, and yield in different tropical sorghum varieties, utilizing a functional phenotyping platform, Plantarray. Cultivars (Kuali, Numbu, Samurai2) were grown under moderate and high salinity, with drought exposure at booting stage. Results showed that Samurai2 had the most significant transpiration reduction under moderate and high salt (36% and 48%) versus Kuali (22% and 42%) and Numbu (19% and 16%). Numbu reduced canopy conductance (25% and 15%) the most compared to Samurai2 (22% and 33%) and Kuali (8% and 35%). In the drought*salinity treatment, transpiration reduction was substantial for Kuali (54% and 57%), Samurai2 (45% and 60%), and Numbu (29% and 26%). Kuali reduced canopy conductance (36% and 53%) more than Numbu (36% and 25%) and Samurai2 (33% and 49%). Biomass, grain yield, and a-100 grain weight declined in all cultivars under both salinity and drought*salinity, and Samurai2 was most significantly affected. WUE$_{biomass}$ significantly increased under drought*salinity. Samurai2 showed reduced WUE$_{grain}$ under drought*salinity, unlike Kuali and Numbu, suggesting complex interactions between water limitation and salinity in tropical sorghum.

**Keywords:** sorghum; Plantarray; HTP; physiological traits; transpiration; yield; abiotic stress; drought tolerance; salinity tolerance; salinity sensitive

## 1. Introduction

Global warming will continue to aggravate various abiotic stresses, such as drought and salinity [1–3]. This poses various challenges to plant breeders and agronomists in developing effective and tailored climate-resilient crop cultivars and adaptive management practices [4,5]. Drought and soil salinization are among the main abiotic stresses and constraints in agriculture—and especially affect semiarid and arid regions, where they put high pressure on the productive land resources [6]. Combined with drought stress, salinity causes crop losses varying from 20% to 50% or more in some major crops, such as in the cereals sorghum, rice, or wheat, which are very important for global food security [7]. To improve agricultural productivity for meeting local food demands, it is therefore important to develop crop adaptation strategies for food production regions that often face drought and salinity stress concurrently.

Progress has been achieved in numerous studies concerning drought and salinity tolerance in various crops. Sorghum is one of the plants with a fairly high tolerance to drought and salinity [8–10]. As a C$_4$ plant, sorghum has high photosynthetic efficiency and

dry matter production. Different sorghum cultivars respond differently to high salt content in the soil with or without water shortages during the growing season, leading to deviating yield penalties [11–13]. Salinity affects the physiological development of sorghum plants by hampering photosynthesis, transpiration, and stomatal conductance [14,15]. Salinity has also been shown to reduce plant height and biomass of sorghum with increasing salt content in the soil [16].

Water stress during sorghum growth has been reported to reduce grain yield and water use efficiency [17]. Exposure to drought in the early growth stages affected morphological characteristics (e.g., reduction in leaf number and plant height) and biomass production of sorghum, even though during the subsequent recovery period, the plant received optimal irrigation until maturity [18]. Progressive water deficit and salinity stress treatments significantly reduce sorghum growth and development by altering various biochemical responses [19].

The physiological mechanisms of combined drought and salinity stress in crops have hardly been investigated [13,15] and, to our knowledge, not at all for sorghum, although numerous scientific studies have revealed the response mechanisms of sorghum to individual drought or salinity stresses [8,10,20,21]. With expected variations in cultivar response to drought and salinity, we hypothesize that, in comparison to salt-resistant cultivars, salt-sensitive sorghum cultivars will exhibit stronger changes in physiological properties and yield components under high-salinity (EC 14 dS m$^{-1}$) and moderate-salinity (EC 7 dS m$^{-1}$) conditions and drought stress. We hypothesize further that, under high-salinity conditions compared to moderate-salinity conditions, salt-sensitive cultivars will experience substantial reductions in physiological performance and yield traits, highlighting a higher susceptibility to the interaction of salt and drought stress.

In this study, we aimed to assess the potential of sorghum as a cropping option in a saline environment that is prone to drought, as is currently the case in the salt-affected coastal lowland farming region of Aceh in Indonesia [22]. To obtain a better insight into the mechanisms of drought and salinity stress occurring concurrently, we designed an experiment under controlled conditions in the greenhouse using the Plantarray system, a high-throughput gravimetric phenotyping (HTP) platform [23]. To this end, we chose three contrasting sorghum cultivars (differing in their sensitivity to salinity and drought) to examine the effect of mid-season drought (i.e., during the reproductive stage, between the booting and soft dough stage) on transpiration rate, canopy conductance, water use efficiency, biomass, and yield traits (grain weight, a-100 grain weight) under moderate (EC 7 dS m$^{-1}$) and high (EC 14 dS m$^{-1}$) salinity levels.

## 2. Materials and Methods

### 2.1. Plant Material and Treatments

Based on a preliminary experiment screening six sorghum cultivars (unpublished) (Supplementary Table S1) for their salt and drought sensitivity, the three Indonesian cultivars Kuali (salt tolerant), Numbu (tolerant to salt and drought), and Samurai2 (sensitive to salt and drought) were selected for this study (see Supplementary Table S1 for further cultivar description).

Plants were sown in seedling trays and transplanted to nursery pots filled with a potting mix of plant compost, peat, and perlite ("Ökohum"). Sixteen days after sowing (DAS), the seedlings were transferred to 5 L pots (sandy loam; one seedling per pot). Before transplanting, soluble salt was added to the soil. EC of 7 dS m$^{-1}$ was obtained by dissolving 4.48 g of NaCl in one liter of demineralized water (assuming that 1 dS m$^{-1}$ = 640 ppm), and 8.96 g was dissolved to obtain EC 14 dS m$^{-1}$ [24]. Then, plants were placed in climate chambers (PGC-105 CLF Plant Climatic GmbH, 1.5 m$^2$, 137 cm height; 12 h of light, temperature 30 °C/22 °C) until mid-tillering (BBCH 23 s), then placed in a semi-controlled greenhouse (temperature 30/22 ± 1.5 °C) until flag leaf emergence (BBCH 41) and then transferred to the Plantarray (see Section 2.2.1). Drought was implemented at the booting stage (BBCH 50 s) through gradual deficit irrigation, i.e., each day, the plant received only

80% of its own previous day's transpiration [25]. Recovery was initiated by reverting to full irrigation as soon as plants showed signs of significant drought stress, i.e., leaf rolling and leaf area reduction [26], and their transpiration had declined to 40% of the potential daily transpiration. The plants were on the Plantarray system from 1 July to 5 August (batch 1) and from 7 August to 6 September 2022 (batch 2) and then placed in the greenhouse again until final harvest.

A summary of the drought and salinity treatments timeline is presented in Figure 1.

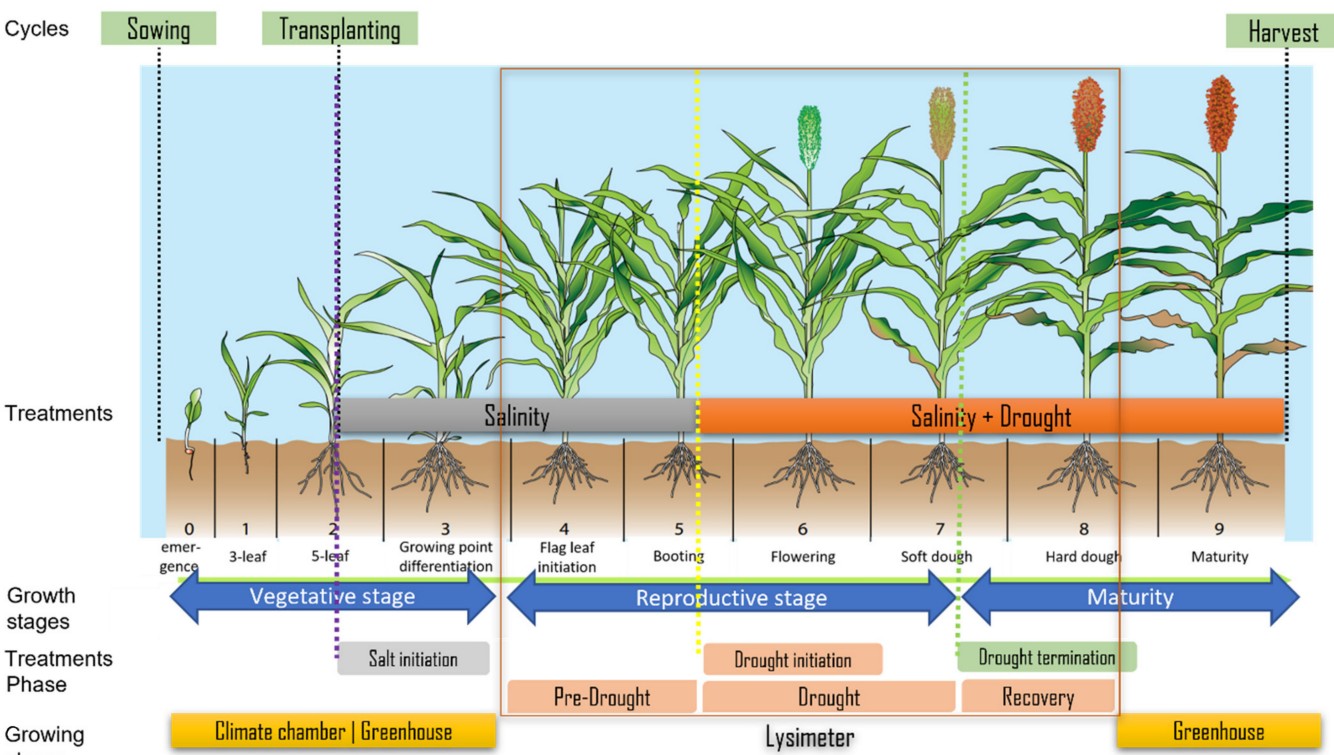

**Figure 1.** Experimental scheme of sorghum drought effect under saline conditions. The black dotted line represents the agricultural cycle timeline. Purple, yellow, and green dotted lines present the salinity and drought treatments applied. Image adapted and modified [27].

### 2.2. Experimental Design

2.2.1. Experimental Set Up Using the High-Throughput Gravimetric Phenotyping (HTP) Platform (Plantarray)

The Plantarray, a high-throughput gravimetric phenotyping (HTP) platform (developed by Plant Ditech; see [23]), was installed in the semi-controlled greenhouse of TROPAGS at the University of Göttingen. The platform incorporates weighing lysimeter units that autonomously, continuously, and simultaneously measure the water flux in the soil–plant continuum for each individual plant (e.g., daily transpiration, transpiration rate, canopy conductance, etc.) (Figure 2. Associated weather stations measure solar radiation, temperature, relative humidity, and vapor pressure deficit (VPD)).

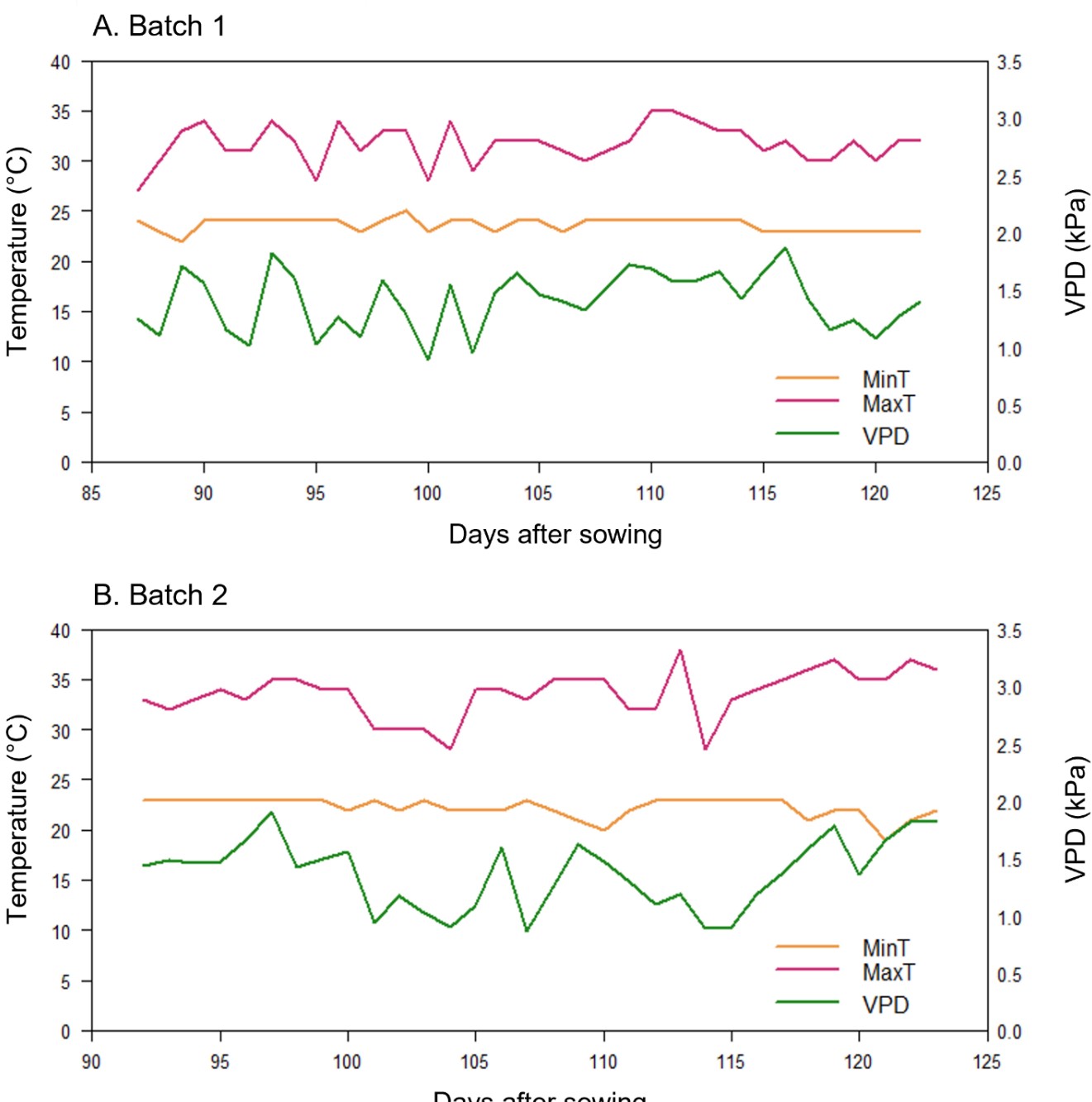

**Figure 2.** Environmental conditions during the experiments in the HTP platform for (**A**) batch 1 (1 July–5 August 2021; 87–122 DAS) and (**B**) batch 2 (7 August–6 September 2021; 92–123 DAS). The red line indicates maximum temperature, the yellow line minimum temperature (°C), and the green shows vapor pressure deficit (VPD, in kPa).

2.2.2. Details on Experimental Conditions

This experiment was conducted in two batches; batch 1 was run on the Plantarray for 35 days and batch 2 for 30 days. Each batch included the three selected cultivars, a control treatment (i.e., well-watered plants and no salt in the soil), and two drought treatments (well watered and drought). In each batch, we tested one salt level (7 dS m$^{-1}$ in batch 1 and 14 dS m$^{-1}$ in batch 2) (Table 1). Each cultivar x treatment combination had four replicates in a randomized complete block design with four blocks laid out as A to D. Each block had nine pots with one replicate for each cultivar x drought treatment (Figure 3). In batch 1, Tmax ranged from 27 to 43 °C and Tmin from 19 to 24 °C. Maximum

VPD ranged from 1.6 to 6 kPa, photoperiod from 12 to 13 h, and maximum PAR recorded between 10.00 and 16.00 h was 832 µmol m$^{-2}$ s$^{-1}$. In batch 2, temperature ranges were 28 to 37 °C for Tmax and 19 to 23 °C for Tmin. Maximum VPD ranged from 1 to 3.6 kPa, photoperiod from 13.5 to 14.9 h, and maximum PAR recorded between 10.00 and 16.00 h was 835 µmol m$^{-2}$ s$^{-1}$ (Figures 2 and S1).

**Table 1.** Experimental design for the different sorghum, salinity, and drought treatments.

| Batch | Cultivars | Treatment | Batch | Cultivars | Treatment |
|-------|-----------|-----------|-------|-----------|-----------|
| 1 | Kuali | Control | 2 | Kuali | Control |
| | | EC 7 well watered | | | EC 14 well watered |
| | | EC 7 drought | | | EC 14 drought |
| | Numbu | Control | | Numbu | Control |
| | | EC 7 well watered | | | EC 14 well watered |
| | | EC 7 drought | | | EC 14 drought |
| | Samurai2 | Control | | Samurai2 | Control |
| | | EC 7 well watered | | | EC 14 well watered |
| | | EC 7 drought | | | EC 14 drought |

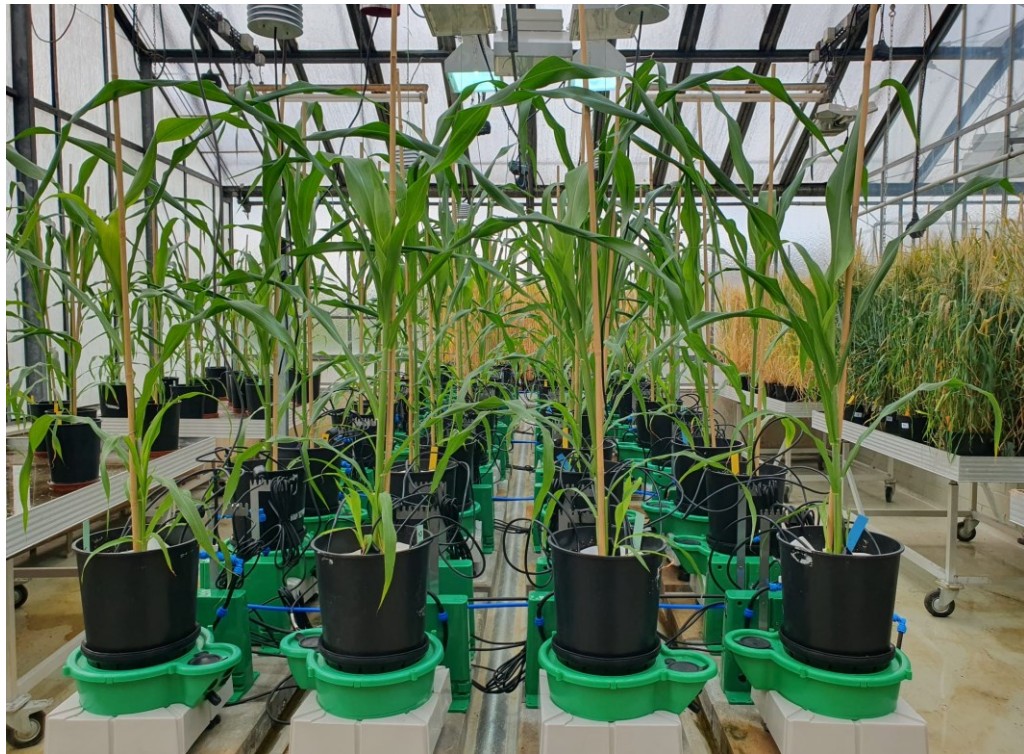

**Figure 3.** Plantarray, a high-throughput functional phenotyping (HTP) platform, was used in this study. Each pot is positioned on a sensitive load cell connected to the control unit.

### 2.3. Data Collection and Statistical Analysis

The system recorded the whole plant's physiological data every 3 min, which could be monitored in real time using the online software "SPAC-Analytics!" (Plant-Ditech, https://spac.plant-ditech.com, accessed on 1 July 2021). The physiological and environmental data presented were collected directly from the software and then analyzed in R. Furthermore, the daily transpiration data were categorized according to different drought-related stages, the pre-drought (at booting stage BBCH50 before drought initiation), drought (on the last day before returning to full irrigation, i.e., recovery), and recovery (one week after recovery was initiated) stages. The cumulative transpiration during the experimental period was calculated as the sum of daily transpiration for all days of the experiment measured by



the Plantarray. The agronomic water use efficiency (WUE) was estimated by dividing the dry biomass and grain yield by the sum of daily transpiration of each batch recorded in the Plantarray.

We recognized the possibility of facing confounding effects due to differences in climatic conditions when testing different salinities in separate experiments. Such confounding effects were especially likely for our case with varying daylength during the summer period. While such separation of experiments was a result of practical constraints due to the limited number of the Plantarray lysimeter units, we mitigated these effects by analyzing and presenting results for each batch separately. The cultivar differences between the batches are therefore relative to their specific environmental conditions. For each batch, we performed a general analysis of variance (ANOVA) to determine the significance of cultivars' response to drought, salinity, and their interactions at $p < 0.05$. All statistical analyses were performed using R studio [28]. We used Levene's test to examine the homogeneity and Shapiro–Wilk test to verify normality of the data. To identify specific differences between cultivars and treatments, as shown in Table 1, we performed Tukey's HSD with a 95% family-wise confidence level.

## 3. Results

### 3.1. Effects of Salinity and Drought Stress on Transpiration

The daily transpiration differed between the three cultivars and also varied among drought treatments (well watered vs. drought) (Figure 4) and salinity levels between batches (see Figure 4A,B).

All cultivars under well-watered and drought treatments showed reduction in daily transpiration at EC 7 dS m$^{-1}$ (Figure 4A).

During drought, the daily transpiration of Kuali was highest (7048 mL), followed by that of Numbu (6555 mL) and Samurai2 (5258 mL), representing 22%, 19%, and 36% of their controls, respectively. During the recovery period, Kuali had the highest daily transpiration at 8984 mL, followed by Numbu (8150 mL) and Samurai2 (6664 mL), which represented 20%, 20%, and 35% of their controls, respectively.

In the salt*drought treatment, no variation was observed in daily transpiration between the cultivars during drought and recovery periods. During the drought period, Kuali recorded 4148 mL ($p < 0.05$) of daily transpiration, followed by Numbu which recorded 5757 mL and Samurai2 which recorded 4566 mL ($p < 0.05$), which showed reductions of 54%, 29%, and 45%, respectively, in comparison to their controls. During the recovery, the highest daily transpiration recorded was 6845 mL in Numbu, followed by that of Samurai2 (5656 mL; $p < 0.05$) and Kuali (5315 mL; $p < 0.05$), showing reductions of 33%, 45%, and 53% compared to their controls, respectively (Table 2).

All cultivars under well-watered and drought treatments demonstrated a more considerable reduction in transpiration in the high-salinity treatment at EC 14 dS m$^{-1}$ (batch 2, Figure 4B). During the drought period, the cultivars' daily transpiration showed that Kuali transpired the most (4195 mL; $p < 0.05$), followed by Numbu (3005 mL) and Samurai2 (2883 mL), corresponding to 42%, 16%, and 48% of the controls' transpiration, respectively. The maximum daily transpiration during the recovery period was recorded in Kuali (7317 mL), followed by 5255 mL in Numbu and 5054 mL in Samurai2 ($p < 0.05$), with reductions of 36%, 14%, and 49% compared to the respective controls.

No differences in daily transpiration between cultivars were noted in the salt–drought treatment during the drought and the recovery phase. Kuali had the greatest transpiration during the drought (3156 mL; $p < 0.05$), followed by Numbu (2642 mL; $p < 0.05$) and Samurai2 (2231 mL), indicating decreases of 57%, 26%, and 60%, respectively, compared to their controls. Kuali had the highest daily transpiration during the recovery period (4647 mL; $p < 0.05$), followed by Numbu (4417 mL) and Samurai2 (3796 mL; $p < 0.05$), all of which had decreased transpiration, by 59%, 14%, and 62%, respectively, compared to their controls (Table 3).

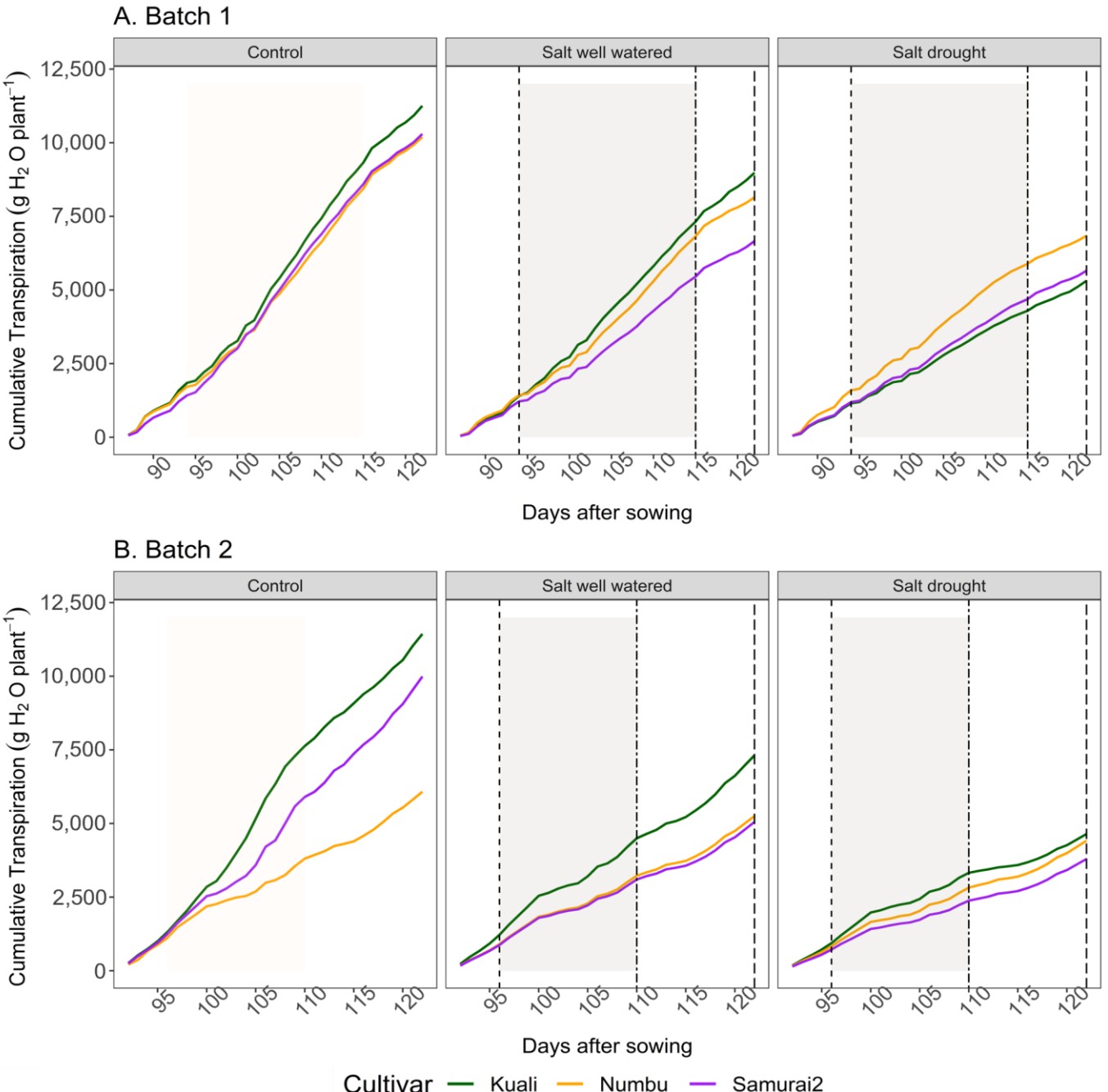

**Figure 4.** Cumulative transpiration of three sorghum cultivars throughout the experimental period for batch 1 (EC 7 dS m$^{-1}$) and batch 2 (EC 14 dS m$^{-1}$) on the Plantarray platform in 2021. Variation in cumulative transpiration of Kuali, Numbu, and Samurai2 as affected by salinity at EC 7 dS m$^{-1}$ (**A**) and EC 14 dS m$^{-1}$ (**B**) under well-watered and drought conditions (i.e., implemented during the booting stage/from drought time point onwards). The dashed line shows the pre-drought end–start day of the drought, the two-dash and shaded lines indicate the drought period, and the long dash suggests the end of the recovery period. The pre-drought period only considered the salt effect, while the drought and recovery period considered the combined salt and drought effect. Control means no salt and drought treatment, salt well watered indicates only salt treatment, and salt drought indicates salt and drought treatment.

**Table 2.** The amount of transpiration (mL per plant) of sorghum cultivars as affected by salinity at EC 7 dS m$^{-1}$ in batch 1 and drought treatment during the pre-drought, drought, and recovery periods.

| Cultivars | Treatment | Cumulative Transpiration per Phase (mL/Plant) | | |
|---|---|---|---|---|
| | | Pre Drought | Drought | Recovery |
| Kuali | Control | 1847 | 8997 a[†] | 11,256 a |
| | Salt well watered | 1407 | 7048 ab | 8984 abc |
| | | (24) | (22) | (20) |
| | Salt–drought | 1153 | 4148 b | 5315 c |
| | | (38) | (54) | (53) |
| Numbu | Control | 1718 | 8142 a | 10,196 ab |
| | Salt well watered | 1432 | 6555 ab | 8150 abc |
| | | (17) | (19) | (20) |
| | Salt–drought | 1594 | 5757 ab | 6845 bc |
| | | (16) | (29) | (33) |
| Samurai2 | Control | 1425 | 8280 a | 10,302 ab |
| | Salt well watered | 1220 | 5258 ab | 6664 bc |
| | | (14) | (36) | (35) |
| | Salt–drought | 1198 | 4566 b | 5656 c |
| | | (16) | (45) | (45) |

[†] Different letters in a different column within the same batch in the same phase (pre-drought, drought, and recovery) indicate significant differences at the $p < 0.05$ level using Tukey's adjusted means comparison test. Values in parenthesis are expressed as a decreased percentage of the control.

**Table 3.** The amount of transpiration (mL per plant) of sorghum cultivars as affected by salinity at EC 14 dS m$^{-1}$ in batch 2 and drought treatment during the pre-drought, drought, and recovery periods.

| Cultivars | Treatment | Cumulative Transpiration per Phase (mL/Plant) | | |
|---|---|---|---|---|
| | | Pre Drought | Drought | Recovery |
| Kuali | Control | 1003 | 7294 a[†] | 11,438 a |
| | Salt well watered | 923 | 4195 bc | 7317 abc |
| | | (8) | (42) | (36) |
| | Salt–drought | 725 | 3156 bc | 4647 c |
| | | (28) | (57) | (59) |
| Numbu | Control | 881 | 3557 bc | 6081 bc |
| | Salt well watered | 698 | 3005 bc | 5255 c |
| | | (21) | (16) | (14) |
| | Salt–drought | 636 | 2642 c | 4417 c |
| | | (28) | (26) | (14) |
| Samurai2 | Control | 952 | 5586 ab | 9994 ab |
| | Salt well watered | 679 | 2883 bc | 5054 c |
| | | (29) | (48) | (49) |
| | Salt–drought | 548 | 2231 c | 3796 c |
| | | (42) | (60) | (62) |

[†] Different letters in a different column within the same batch in the same phase (pre-drought, drought, and recovery) indicate significant differences at the $p < 0.05$ level using Tukey's adjusted means comparison test. Values in parenthesis are expressed as a decreased percentage of the control.

At EC 7 dS m$^{-1}$, in the well-watered treatment, canopy conductance was significantly different between Kuali and Samurai2 but not compared with Numbu. Within the cultivar, Kuali had 36.27 g/min of canopy conductance, Numbu 30.31 g/min ($p < 0.05$), and Samurai2 25.88 g/min, which represents an increase of 8% and a decrease of 25% and 22%, respectively, compared to their controls. In the salt and drought treatment, we discovered that Kuali was significantly different to Numbu but not to Samurai2. The highest canopy conductance was in Numbu (30.42 g/min; $p < 0.05$), followed by Kuali (21.27 g/min; $p < 0.05$) and Samurai2 (22.46 g/min), representing a decrease of 25%, 36%, and 33%, respectively, compared to their control treatments (Figure 5A).

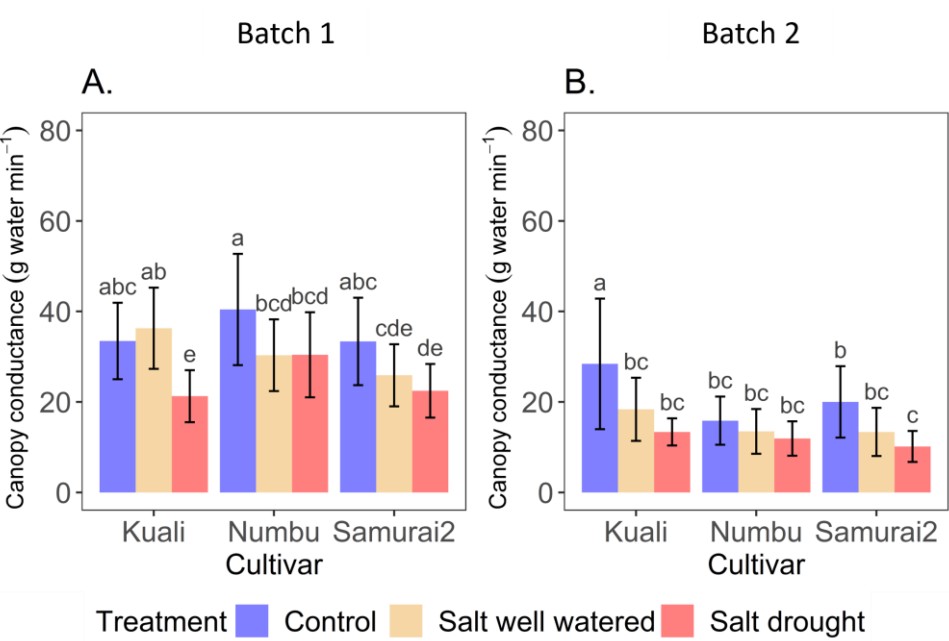

**Figure 5.** Average of daily canopy conductance of three sorghum cultivars from the first to last day of the drought period for batch 1 (EC 7 dS m$^{-1}$) (**A**) and batch 2 (EC 14 dS m$^{-1}$) (**B**) on the Plantarray platform in 2021. Letters indicate significant differences at the $p < 0.05$ level using Tukey's adjusted means comparison test, and bars represent the standard deviation.

At EC 14 dS m$^{-1}$, in the well-watered treatment, no difference was found between the cultivars. However, we recorded the highest canopy conductance in Kuali (18.35 g/min; $p < 0.05$), followed by Numbu (13.50 g/min) and Samurai2 (13.36 g/min), representing a 35%, 15%, and 33% reduction compared to their controls (Figure 5B). Under combined salt and drought, there also was no difference between the cultivars. When measured, Kuali had the highest canopy conductance (13.37 g/min; $p < 0.05$), followed by Numbu (11.92 g/min) and Samurai2 (10.15 g/min; $p < 0.05$), reduced by 53%, 25%, and 49%, respectively, compared to their controls (Figure 5B).

### 3.2. Effect of Salinity and Drought on Water Use Efficiency

Despite the absence of substantial difference between the cultivars, the WUE$_{biomass}$ increased under salt and drought treatments at both salinity levels between batches (Figure 6).

Under the moderate salinity level at EC 7 dS m$^{-1}$, WUE$_{biomass}$ of Numbu and Samurai2 was at 0.10 g for both, and Kuali showed 0.09 g. It represents a 16% and 11% increase and 16% decrease compared to their controls, individually. Under combined salt and drought, Kuali showed the highest WUE$_{biomass}$ at 0.16 g, and it was 0.12 g in Samurai2 and 0.10 g in Numbu, implying an increase of 51%, 31%, and 25%, respectively, compared to control plants (Figure 6A). At the same salinity level (moderate), the WUE$_{grain}$ significantly differed between the Kuali and Numbu but not for Samurai2. The highest WUE$_{grain}$ was recorded in Numbu (0.011 g), followed by Samurai2 (0.005 g) and Kuali (0.002 g), representing an increase of 39% and a decrease of 19% and 49% compared to their controls, respectively. Under combined salt and drought, the highest WUE$_{grain}$ was in Numbu (0.010 g), followed by Samurai2 (0.006 g) and Kuali (0.005 g), representing an increase of 20% and 2% and a 27% decrease, respectively, compared to their controls (Figure 6B).

Under the high salinity level at EC 14 dS m$^{-1}$, WUE$_{biomass}$ of Numbu was 0.08 g, 0.06 g for Samurai2, and 0.05 g for Kuali, indicating a 9%, 12%, and 5% increase compared to controls, individually. Under salinity and drought, WUE$_{biomass}$ of Numbu and Samurai2 was 0.09 g for both, followed by 0.08 g for Kuali, representing a 25%, 61%, and 75% increase, respectively, compared to controls (Figure 6C). At this high salinity, no variation was found between all cultivars in WUE$_{grain}$. The highest WUE$_{grain}$ was recorded in Numbu (0.017 g),

followed by Samurai2 (0.008 g) and Kuali (0.008 g), which showed a 12% and 75% decrease and a 14% increase compared to their controls, respectively. Under combined salt and drought, the highest $WUE_{grain}$ was in Numbu (0.022 g), followed by Samurai2 (0.013 g) and Kuali (0.010 g), which showed an increase of 20%, a 1% decrease, and a 33% increase compared to their controls, individually (Figure 6D).

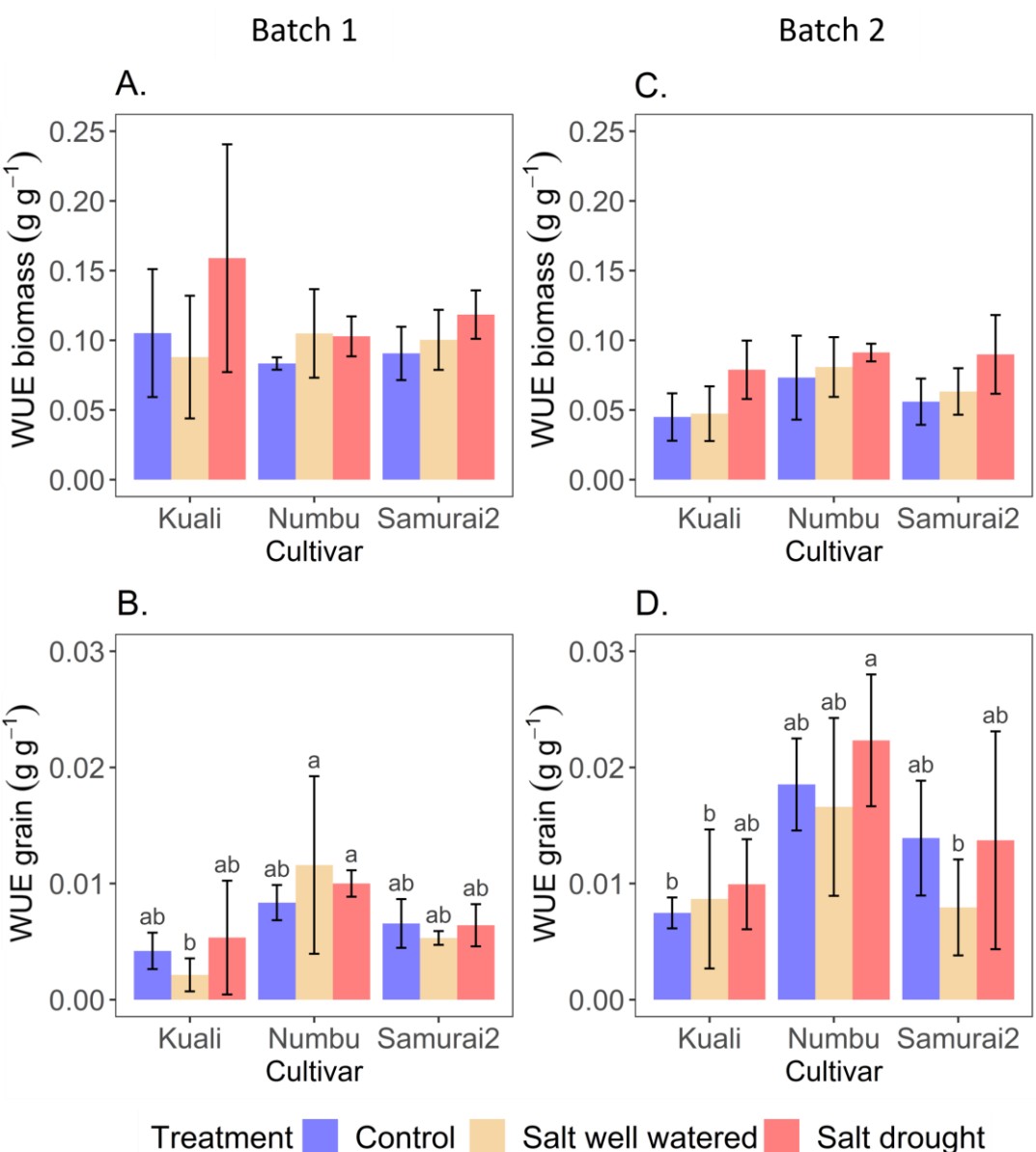

**Figure 6.** Water use efficiencies related to biomass (Figure 4A,B) and to grain (Figure 4C,D) of the three sorghum cultivars in batch 1 (EC 7 dS m$^{-1}$) (**A,C**) and batch 2 (EC 14 dS m$^{-1}$) (**B,D**). Letters indicate significant differences at the $p < 0.05$ level using Tukey's adjusted means comparison test, and bars represent the standard deviation. Graphs with no letters on the bars indicate no significant differences at $p < 0.05$.

### 3.3. Effect of Salinity and Drought on Yield Component

Salinity and drought negatively affected the two main productivity traits of dry biomass and yield (Figure 7). When affected by high salt at EC 7 dS m$^{-1}$ in well-watered conditions, we observed no variation in biomass between the three cultivars. Biomass of Numbu was recorded at 245 g, of Samurai2 at 210 g, and of Kuali at 205 g, indicating reductions of 10%, 9%, and 19%, respectively, compared to control plants. Under salt and

drought, Numbu, Kuali, and Samurai2 biomass was recorded as 244 g, 224 g, and 216 g, respectively, representing an 11% drop for both Numbu and Kuali and a 7% reduction for Samurai2 versus the control treatments (Figure 7A).

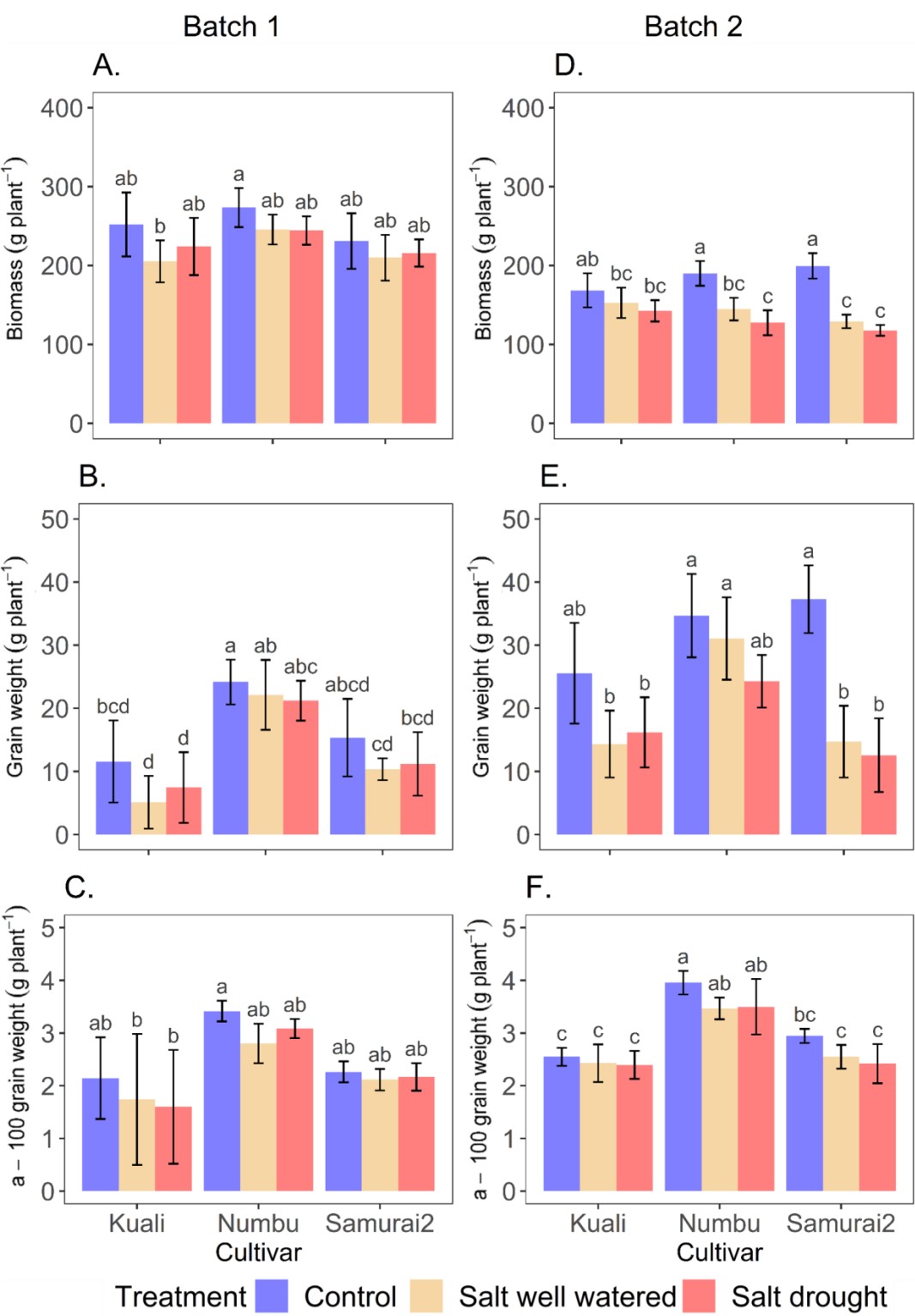

**Figure 7.** Mean of biomass (**A,B**), grain weight (**C,D**), and a-100 grain weight (**E,F**) of each sorghum cultivar and treatment under two salinity levels; (**A**–**C**) EC 7 dS m$^{-1}$, (**D**–**F**) EC 14 dS m$^{-1}$, with bars representing standard deviation and letter indicate significant differences at the $p < 0.05$ level using Tukey's adjusted means comparison test.

At the same salt level in the well-watered treatment, significant variation in grain weight was shown between Kuali and Numbu but not for Samurai2. The treatments within the cultivar showed the highest grain weight was recorded in Numbu (22.11 g), followed by Samurai2 (10.34 g) and Kuali (5.11 g), resulting in an 8%, 33%, and 56% reduction compared to the controls. In the salt–drought treatment, we observed significant variation between Kuali and Numbu but not in Samurai2. The highest grain weight was found in Numbu (21.20 g), followed by Samurai2 (11.20 g) and Kuali (7.43 g), referring to a decrease of 12%, 27%, and 36% compared to control plants (Figure 7B).

No significant difference was found between all cultivars in a-100 grain weight in the salt–well-watered treatment at this level. The treatment within the cultivar showed that Numbu had the highest a-100 grain weight at 2.80 g, followed by Samurai2 (2.11 g) and Kuali (1.78 g), indicating a reduction of 18%, 7%, and 19% in weight compared to control plants. In the salt–drought treatment, no significant difference was found between cultivars. The treatment within the cultivar recorded an a-100 grain weight in Numbu of 3.08 g, in Samurai2 an a-100 grain weight of 2.16 g, and in Kuali an a-100 grain weight of 1.59 g, indicating a 10%, 4%, and 25% reduction compared to control plants (Figure 7C).

Under the EC 14 dS m$^{-1}$ well-watered treatment, there was also no variation in biomass between the cultivars. However, we recorded biomass in Kuali at 153 g, in Numbu at 145 g ($p < 0.05$), and in Samurai2 at 129 g ($p < 0.05$) per plant, corresponding to a reduction of 9%, 24%, and 35%, respectively, compared to the controls. Under combined salt and drought, a biomass of 143 g was observed for Kuali, for Numbu 127 g ($p < 0.05$), and for Samurai2 118 g ($p < 0.05$), indicating a 15%, 33%, and 41% reduction compared to control treatments (Figure 7D).

At the similar salinity level in the well-watered treatment, Kuali was significantly different from Numbu but not Samurai2 in grain weight. The highest grain weight was found in Numbu (31.06 g), followed by Samurai2 (14.70 g) and Kuali (14.34 g), indicating a 10%, 61%, and 44% reduction compared to their control treatments. Under combined salt and drought, Numbu had the highest grain weight at 24.28 g, followed by Kuali (16.19 g) and Samurai2 (12.56 g; $p < 0.05$), representing a 30%, 37%, and 66% reduction compared to the control treatments, respectively (Figure 7E).

a-100 grain under the same salt showed significant differences between Kuali and Numbu but not for Samurai2. The treatments within the cultivar showed that the highest weight was found in Numbu (3.46 g), followed by Samurai2 (2.54 g) and Kuali (2.42 g), representing a 12%, 13%, and 5% decrease, respectively, compared to the control treatments. Significant differences in a-100 grain between Kuali and Numbu were shown during salt and drought but not for Samurai2. The highest was recorded in Numbu (3.49 g), followed by Samurai2 (2.41 g) and Kuali (2.39 g), indicating a 12%, 18%, and 6% decrease compared to control plants (Figure 7F).

## 4. Discussion

This study, using the Plantarray platform, focused on how drought under different saline environments (moderate at EC 7 dS m$^{-1}$—batch 1 and high salinity levels at EC 14 dS m$^{-1}$—batch 2) affects physiological characteristics, water use efficiency, biomass, and yield components of different sorghum cultivars. Our findings support the initial hypothesis, i.e., that for all three cultivars, Kuali (a salt-tolerant cultivar), Numbu (a salt–drought-tolerant cultivar), and Samurai2 (a salt–drought-sensitive cultivar), drought (water deficit), particularly during the booting stage, intensifies the impact of salinity on the transpiration, canopy conductance, and WUE of sorghum. The biomass and yield-related components, such as grain and a-100 grain weight, were further depressed by the drought. In the following, we discuss in detail the implications of drought stress and salinity for sorghum physiological characteristics and yield components, starting with transpiration and canopy conductance (Section 4.1), turning to water use efficiency in Section 4.2 and yield components in Section 4.3.

### 4.1. Transpiration and Canopy Conductance in Response to Salinity and Salinity + Drought

The difference in transpiration responses across the cultivars in the control treatment in both batches is mostly attributable to variations in environmental variables (e.g., temperature, humidity, and light intensity) during the growing periods. Sorghum has been shown to be a photoperiod-sensitive crop categorized as a short-day plant [29,30]. According to the literature, the optimum daylength for sorghum ranges from 12.5 to 13.5 h [31].

Exceeding the upper threshold value (e.g., 13.5 to 14.9 h, as observed in batch 2) can increase the number of initiated leaves and delay full flag leaf emergence. These alterations in characteristics are likely to result in increased transpiration [31–33]. However, longer daylength and lower VPD and PAR levels decreased the transpiration; moreover, the phenological development was slower in batch 1, where cultivars were mostly in early and late flowering stages, while in batch 2, they had already advanced to the late milk stage. Sorghum germplasm that originates near the equator exhibits high sensitivity to even minor changes in daylength, with variations as slight as < 15 min significantly affecting plant growth and development [33]. The three cultivars tested in our study all originated from within the equatorial zone (Supplementary Table S1). Evaluating germplasm accessions from specific geographic regions will be crucial for identifying adaptive traits to further enhance the tolerance of sorghum to drought and salinity [34]. Such information will be invaluable for breeders when selecting and developing resilient sorghum cultivars [18].

The adverse effect of salinity and drought stress on physiological traits, as found in this study, is quite typical and in line with the literature on other tropical cultivars [11,20,24]. A study reported that salinity causes toxic ion accumulation in the leaves, which reduces photosynthetic rate, transpiration rate, and canopy conductance when plants are exposed to salinity levels above the tolerance threshold [35]. Under a moderate salinity of EC 7 dS m$^{-1}$, as implemented in our experiment, canopy conductance did significantly vary among Kuali, Numbu, and Samurai2. However, the cumulative transpiration was not significantly reduced in all cultivars (Figure 4A and Table 2). Such a moderate salinity level is assumed to be within the tolerance margin of the cultivars tested in this study. For example, [36] reported that, for two sorghum cultivars irrigated with saline water at 6.8 dS m$^{-1}$ and grown in the western United States, yield was not affected. The effect of moderate salinity only became pronounced for Kuali and Samurai2 when drought was introduced, yet salt-tolerant Numbu was not affected. Furthermore, salt significantly reduced canopy conductance, i.e., by 25% for Numbu, while for Kuali and Samurai2, the decrease was more than 30% (Figure 5A). In sorghum, leaf stomatal conductance tends to be diminished when salinity levels fall below 9 dS m$^{-1}$ [37]. Interestingly, salt-tolerant Kuali reduced cumulative transpiration slightly more than salt–drought-sensitive Samurai2 (Section 2.1) under salinity ($p > 0.05$ drought). The combined impacts of salinity and drought on functional traits in crops largely depend on the duration of the growing season and the phenological phase, as was shown for maize and potato [38]. Most likely for Kuali, the presence of drought during the early generative growth stage exacerbated the effects of salt, as the transpiration reduced about one third more under the concurrent (salinity; $p > 0.05$ drought) stress event (Table 2).

At the moderate salinity level, especially salt-tolerant cultivars such as Numbu can extract more water and nutrients from salinized soil than the other two salt-sensitive cultivars. The salt concentration is considered less harmful at this level; therefore, soil water absorption is enabled even under light-to-moderate drought. As a result, the impact on a salt-tolerant cultivar like Numbu was much less severe than on Kuali and Samurai2. In this case, the two combined abiotic stresses might not affect transpiration as long as water uptake is reduced by decreased osmotic potential and no toxic effects or nutrient imbalances occur. Defense mechanisms such as an osmotic adjustment in dry conditions are found in sorghum cultivars that are tolerant to stress [39].

At the higher salinity level, i.e., at EC 14 dS m$^{-1}$, transpiration was significantly decreased in the most salt-sensitive cultivar, Samurai2, but not in Kuali and Numbu (Figure 4B and Table 3). Although transpiration in Kuali was not significantly decreased,

for canopy conductance, a decline of 35% was recorded. As a cultivar susceptible to salt, Samurai2 was significantly impacted by the high salt accumulation in the soil. The other two cultivars, Kuali and Numbu, showed medium and high tolerance to the high salinity level, respectively. Salinity-sensitive genotypes of sorghum might experience greater reductions in leaf transpiration compared to tolerant genotypes [20], corroborating the data and insights obtained from our study. It has been shown that prolonged salinity stress reduces some gas exchange parameters, including transpiration, stomatal conductance, and net photosynthetic rate, of sorghum [24].

During drought stress, there was a noticeable decrease in transpiration and canopy conductance, particularly in Samurai2 and Kuali. Numbu, however, showed a similar water uptake pattern as under salinity conditions only. For salt-sensitive cultivars such as Samurai2, the extreme $Na^+$ and $Cl^-$ accumulation in the soil affects the plant's ability to extract available water and disrupts the water movement in the plant. The strong effects on transpiration in Kuali showed that this cultivar was more susceptible to drought in saline soil. High salt concentrations limit the growth and development of crops in multiple ways. A well-known effect is that phytotoxicity and osmotic imbalance contribute to physiological drought [40]. Drought exposure ultimately exacerbates the salinity impact [41], as we found for Kuali and Samurai2. An increase in salinity and drought stress caused a decrease in some physiological processes (i.e., transpiration, stomatal conductance, net photosynthetic activity) [42] and characteristics in sorghum and also in *Brassica oleracea* [43].

### 4.2. Water Use Efficiency in Response to Salinity and Salinity + Drought

$WUE_{biomass}$ increased under salt only at EC 7 dS $m^{-1}$ in Numbu and Samurai2 but not in Kuali. $WUE_{grain}$ also increased in salt-tolerant cultivar Numbu but not in Kuali and Samurai2 under salinity stress. The combination of salinity and drought increased the $WUE_{biomass}$ and $WUE_{grain}$ in all cultivars but to different degrees. A study revealed that sorghum exposed to less saline conditions has a higher WUE [44]. Water stress and mild salt stress have been observed to increase $WUE_{grain}$ in crops, e.g., maize [45].

Furthermore, at higher salinity levels (EC 14 dS $m^{-1}$), only $WUE_{biomass}$ was increased by salt in all cultivars. Under drought and salinity stress, Kuali increased the $WUE_{biomass}$ more than Samurai2 and Numbu. The $WUE_{grain}$ under high salinity varied among the cultivars—salt increased the WUE in Kuali but not in Numbu and Samurai2. When combined with drought, our results suggest that salt increases $WUE_{grain}$. Salt treatment strongly influences the WUE, e.g., in quinoa, the WUE increased clearly as salt content increased [46]. Drought reduced transpiration as salinity rose, conserving WUE at high salinity levels and enabling sorghum to withstand high salinity (10 dS $m^{-1}$) [42].

### 4.3. Yield Components Variation in Response to Salinity and Salinity + Drought

Although transpiration decreased equally in Kuali and Numbu (Table 2), the reductions in biomass production and the yield components (e.g., grain and a-100 grain weight) were more significant in Kuali than in Numbu with the moderate salt content (Figure 6A).

It is possible that assimilation and conversion into dry matter for Kuali were less efficient than for Numbu. For instance, it might be that, due to differences in leaf area development and the pre- and post-anthesis growth durations, assimilates in Numbu were utilized more efficiently for grain filling. More generally, it has been shown that sorghum has the potential to continue growing as the dilution of $Na^+$ in leaves enables the plant to maintain stomatal opening and continue to produce enough carbon to support the overall plant [42,47]. A study found a reduction of up to 30% in sorghum grain yield when the crop was irrigated with salt water at EC 6 dS $m^{-1}$ [44]. A similar decrease was found in our sensitive cultivar, Samurai2; however, the grain reduction was not significantly different when subjected to the moderate salt level, as also found in [44].

Under the high-salinity treatment, Kuali responded inconsistently to drought regarding biomass and grain yield. We anticipated that the effect would be comparable to that of combined stress under low salinity. For a salt-sensitive cultivar, as anticipated for Samu-

rai2, it would not be surprising to find that salt stress and the presence of drought have a significant impact on photosynthesis and physiological activity, increasing the effect by considerably reducing transpiration rate and, consequently, agronomic yield [48]. Although some genotypes are tolerant to a moderate salinity level during the vegetative phase, which is indicated by more biomass gain, for sensitive cultivars, salinity results in yield penalties (e.g., number of panicles and a-100 grain weight) [49].

For Samurai2 and Numbu, the higher salt content in the soil caused a significant loss in biomass but not in grain yield. Numbu seems to reduce biomass to maintain resources such as water and nutrients for yield. Strategies to reduce biomass production to allow grain development and production are essential mechanisms for addressing salinity exposure [20]. The primary response of crops to salinity is restricted transport of salt to the shoot, where then relatively favorable water conditions are maintained to synthesize organic solutes, which protects them from the adverse effects of salinity [50]. The better ability of Numbu to manage water loss as compared to the other two cultivars affects its yield traits and biomass less negatively when exposed to drought in a saline environment. Sorghum has very good potential for adaptation to drought and salinity. Grain yield and biomass reductions are considered fairly negligible at low or medium salinity levels and under water deficit [8].

Differences in yield and physiological response to combined drought and salinity stress among crop cultivars were also observed in barley, where tolerant cultivars were less affected regarding yield and biomass than the sensitive ones [11]. The ability to maintain biomass growth under salinity is considered an indicator of salinity tolerance in sorghum genotypes [20]; however, sorghum is sensitive to water stress during the flowering stage, which then eventually leads to yield penalties [51].

## 5. Conclusions

We used a high-throughput functional phenotyping platform (Plantarray) to assess the response of three tropical sorghum cultivars to salinity and combined salinity and drought stress. The three cultivars, originating from Indonesia, showed markedly different responses regarding physiological traits (i.e., transpiration, canopy conductance, and WUE), biomass, and yield components (i.e., grain and a-100 grain weight).

The data revealed that moderate salinity induced a small reduction in transpiration as well as in canopy conductance in all cultivars. A more significant adverse effect was reduced transpiration when salinity was combined with drought stress in Kuali and Samurai2—but not in Numbu. WUE$_{biomass}$ was not substantially affected by salinity only, nor by combining salinity with drought stress. This was true for all cultivars.

The high salinity levels resulted in decreased transpiration, canopy conductance, and water use efficiency across all cultivars. The most pronounced impact was observed in the transpiration of Kuali and Samurai, while Numbu remained less affected. Biomass and grain weight also experienced reductions due to salinity, both independently and in combination with drought, except for in Numbu.

These findings relating to the interactions of moderate and high salinity with drought are of vital interest for improving crop production in areas that are prone to both salinity and drought. The research revealed that the higher the salinity, the greater the yield penalty. Our study with the Plantarray system has been pivotal for expanding analyses and designing field experiments to assess sorghum's responses to drought and salinity. Future research should involve diverse sorghum cultivars, combining greenhouse pre-testing with the Plantarray system for comprehensive data collection to support practical decision-making in drought- and salinity-prone tropical lowland agriculture. These findings emphasize the complex interplay between water limitation and salinity in tropical sorghum, highlighting the need for comprehensive management strategies. Encouraging the adoption of robust sorghum varieties can enhance productivity and mitigate yield losses. Collaborative efforts between farmers and breeders can further advance the development of sustainable varieties, benefiting all stakeholders.

**Supplementary Materials:** The following supporting information can be downloaded at https://www.mdpi.com/article/10.3390/agronomy13112788/s1, Table S1: Characteristics of all sorghum cultivars assessed for drought and salinity; Figure S1: Maximum Photosynthetically Active Radiation (PAR) during the experiments in the HTP platform for (**A**) batch 1 (1 July–5 August 2021; 87–122 DAS) and (**B**) batch 2 (7 August–6 September 2021; 92–123 DAS), which can be downloaded at https://zenodo.org/records/10084192.

**Author Contributions:** All authors conceived the study and were involved in its design. E.S.D.: writing—original draft preparation. E.S.D., I.A., G.B.-M., M.A. and R.P.R.: writing—review and editing. I.A., G.B.-M. and E.S.D.: visualization. R.P.R.: coordination of internal reviews and supervision. All authors have read and agreed to the published version of the manuscript.

**Funding:** E.S.D received funding from the Indonesia Endowment Fund for Education (Lembaga Pengelola Dana Pendidikan—LPDP), Ministry of Finance, Republic of Indonesia (Grant No. S-1273/LPDP.3/2018). E.S.D and I.A funded by the Division of TROPAGS, Department of Crop Sciences, University of Göttingen, Germany. M.A and R.P.R receive funding from BRACE (Barley Responses and Adaptation to Changing Environments) carried out under the ERA-NET Cofund SusCrop being part of FACCE-JPI (Grant No. 771134).

**Data Availability Statement:** The data presented in this study are available on request from the corresponding author.

**Acknowledgments:** This study was supported by the Division TROPAGS, Department of Crop Sciences, University of Göttingen, Germany, and Universitas Malikussaleh, Indonesia. We acknowledge support by the Open Access Publication Funds of the Göttingen University.

**Conflicts of Interest:** The authors declare no conflict of interest.

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
