# Peer review of "Agronomic and Physiological Traits Response of Three Tropical Sorghum (Sorghum bicolor L.) Cultivars to Drought and Salinity"

_agronomy, doi:10.3390/agronomy13112788_

Round 1

Reviewer 1 Report

Comments and Suggestions for Authors

Dear authors and editor,

This work addresses the evaluation of three contrasting sorghum cultivars under saline and saline*drought conditions as compared to controls in what regards to their transpirative and yielding performance. 

Overall quality of work is good, literature and methods are appropriate. In my opinion, the novelty is relatively low and the importance of the work (e.g. advice to breeders/farmers) is not sufficiently clear, therefore, this must be highlighted in the MS. My main concerns are the following:

- Regarding the experimental design, authors compared control treatments to salinity and salinity+drought. Therefore, authors cannot discuss/conclude on the effect of water stress alone. Thus, I suggest to reformulate/re-write all the assertions in the MS regarding "drought effect" and change to "drought*salinity effect", for instance. In addition, and in relation to the chosen design, I do not believe that the equation (L168) of the proposed model is valid and should be rewritten.

- It appears that differences in transpirative and yielding performance were big between batches, even higher that between treatments/cultivars. I think this is not enough argued thoughout the Results and Discussion, and needs to be addressed.   

- As for the results section, I think it is clear and well organized, but some paragraphs could be rewritten to improve readability, avoid repetitions and, above all, make it clearer when certain differences are significant and when they are only non-significant trends.

- How was the biomass calculated/measured? If it is explained I missed it, is it on a dry or fresh basis? I think it is too high to be on a dry basis, especially considering it is a greenhouse experiment.

* Besides, some few comments in attachement.

Author Response

Dear Reviewer,

We would like to extend our sincere appreciation for taking the time to review our manuscript and for considering our response, as attached. Your valuable insights and feedback are greatly appreciated and will undoubtedly contribute to the improvement of our work. Thank you for your dedication to the peer review process.

Warm regards,

Authors

Reviewer 2 Report

Comments and Suggestions for Authors

The study conducted by Dewi et al., described the effect of combined salt*drought stress of sorghum plants development. The approach of the authors for the development and relevance of the subject is adequate, the information provided is orderly and coherent but not too much to be published. I think that tose fundings are still preliminary as authors studied only th effect of those stresses on the biomoass, Water use efficiency, grain weight and 100 grains weight. Thus, i have some remarks to highlight those fundings.

 1-     Can you add some other physiological traits such as number of leaves, roots length/weight, number of roots……

 2-     Are the yields affected by salinity, drought and the combined treatments  in the studied varieties ?

 3-     It would be nice to add some molecular analysis using RT-PCR (studying some genes implicated in plant response to abiotic stress essentially salt and drought stress such as antioxydant system, HKT antiporters, MAP Kinases……

Author Response

Dear Reviewer,

We would like to extend our sincere appreciation for reviewing our manuscript and considering our response, as attached. Your valuable insights and feedback are instrumental in improving the quality and rigor of our research. Your dedication to the peer-review process is greatly appreciated.

Thank you once again for your contribution to the advancement of our work.

Best regards,

Authors

Reviewer 3 Report

Comments and Suggestions for Authors

The study aims to evaluate the yield and physiological responses of sorghum cultivars to drought and salinity. Despite its significant potential for yielding valuable insights, a critical drawback in the experimental design hinders the ability to examine the results and draw reliable inferences.

The limitations in the experimental design prevent a proper analysis of the salinity factor levels and their interactions with water availability.

MAJOR QUESTIONS:

- The study lacks any mention of the solar radiation conditions prevailing during the experimental periods. Can you please provide a description of these conditions?

- Could you please provide information regarding the installation of the Plantarray system? Specifically, was the Plantarray system installed in a greenhouse, and were any environmental control measures implemented during the course of the study?

- Line 107-108 indicates that “The recovery of plants was done by reverting to full irrigation as soon as their transpiration had declined to 40% of their potential daily transpiration”. However, according to Line 162 “The drought’s end was determined based on severe drought symptoms (…)”. Can you please explain?

- Line 168 - Can you provide information about the specific levels of the factor salinity used in the mixed model presented?

- Line 168 - Can you provide information about the specific levels of the factor water regime used in the mixed model presented?

- Line 444 - Can this conclusion be drawn from the results?

- Line 446 states that “Our research revealed that the higher the salinity, the greater the yield penalty”. Do the results support this statement?

Author Response

Dear Reviewer,

We extend our sincere thanks for taking the time to review our manuscript and for carefully considering our response, as attached. Your valuable insights and feedback are greatly appreciated and will undoubtedly contribute to the improvement of our work.

Best regards,

Authors

Reviewer 4 Report

Comments and Suggestions for Authors

It is interesting study and worth to be investigated in further research.

Please see my feedback:

-Methodology part has long introduction, can be reduced

-Too many details in result part, can be simplified

-Formulas in statistics part can be removed

Comments on the Quality of English Language

Needs proofreading for English

Author Response

Dear Reviewer,

We appreciate your positive feedback on our manuscript, "Agronomic and physiological traits response of three tropical sorghum (Sorghum bicolor, L.) cultivars to drought and salinity." 

Your comments have been invaluable in improving our work, and we have incorporated your suggestions as detailed in our responses. 

Thank you for your valuable input.

Sincerely,

Elvira S Dewi

Round 2

Reviewer 2 Report

Comments and Suggestions for Authors

Respected authors,

Thank you very much for this ameliorated version of the manuscript 

Best regards

Author Response

Dear Reviewer,

We greatly appreciate your positive feedback and valuable comments on our manuscript, "Agronomic and physiological traits response of three tropical sorghum (Sorghum bicolor, L.) cultivars to drought and salinity." Your insights have significantly improved our work, and we have addressed your suggestions in our responses.

Thank you for your valuable input.

Sincerely,

Elvira S Dewi

Reviewer 3 Report

Comments and Suggestions for Authors

Dear authors thank you for considering my comments on the review of your article.

I understand that the limited number of experimental units limited your ability to thoroughly analyse the intricate interactions between different salinity factor levels and their effects in different water availability scenarios. However, a major drawback remains in the experimental design used. The fact that the two salinities were tested in different experiments meant that some of the climatic variables could have had a significant influence on the results obtained (remember that the mixed model does not take into account environmental variables such as air temperature, VPD and photoperiod).

The fact that there were differences in the number of hours of light (batch 1: 12 to 13 h and batch 2: 13.5 to 14.9 h), the maximum VPD (batch 1: 1.6 to 6 kPa and batch 2: 1 to 3.6 kPa) and the maximum air temperature (batch 1: 27 to 43 ºC and batch 2: 28 to 37 ºC) between the two experiments makes it difficult to analyse the data on the influence of salinity and drought on production.

Best wishes

Author Response

Dear Editor, 

We appreciate your insightful review of our manuscript, "Agronomic and physiological traits response of three tropical sorghum (Sorghum bicolor, L.) cultivars to drought and salinity." 

Your feedback was constructive, and we have incorporated your suggestions as outlined in our responses to your review. 

Thank you for your valuable input.

Sincerely,

Elvira S Dewi
